# The Dynamics of Sugar Maize (*Zea mays saccharata* Sturt.) Infestation of Field Pansy (*Viola arvensis*)

**DOI:** 10.3390/plants12203581

**Published:** 2023-10-15

**Authors:** Hubert Waligóra, Leszek Majchrzak, Bogna Zawieja, Robert Idziak, Piotr Szulc

**Affiliations:** 1Department of Agronomy, Poznan University of Life Sciences, 60-632 Poznan, Poland; hubert.waligora@up.poznan.pl (H.W.); leszek.majchrzak@up.poznan.pl (L.M.); robert.idziak@up.poznan.pl (R.I.); 2Department of Mathematical and Statistical Methods, Poznan University of Life Sciences, 60-637 Poznan, Poland; bogna.zawieja@up.poznan.pl

**Keywords:** sugar maize, field pansy, weeds, previous crop

## Abstract

Field pansy infestation can lead to a decrease in the species diversity of plant communities and to the disappearance of other species. Field pansy infestation is fairly common in many crops, including maize. Understanding the ecology and management strategies for field pansy in maize is essential for effective weed control. This research into sugar maize was conducted from 1992 to 2019 in the Research and Education Center Gorzyń, Złotniki branch, which belongs to the Poznań University of Life Sciences. The assessment of weed infestation was carried out in experiments that focused on chemical weed control in maize. The experiments were established as single-factor randomized block designs with four field replications. The aim of the study was to evaluate dynamic changes in the status and the degree of field pansy infestation in sugar maize that was cultivated after various other crops in the Wielkopolska region, with a focus on weather conditions. The results indicated that the probability of field pansy individuals occurring among the total number of weeds was highest when maize was cultivated after wheat, but the probability of such infestation did not significantly differ when maize was sown in a crop rotation after winter triticale.

## 1. Introduction

The biodiversity of weed communities is determined by habitat factors, resulting in the occurrence of nitrophilous and alkaline loving-species such as *Amaranthus retroflexus* L., *Matricaria inodora* L., and *Thlaspi arvense* L. in maize cultivated on fertile soil complexes and species such as *Anthemis arvensis* L. and *Viola arvensis* Murray (field pansy) in maize cultivated on soil complexes associated with rye [1,2]. The primary source of weed infestation is the soil’s seed bank, which consists of accumulated weed diaspores. Frequent simplification of cultivation practices and the monoculture of maize can alter weed infestation levels resulting from cultivation practices based on plowing and crop rotation [3]. Infestation of *V. arvensis* can lead to a decrease in the species diversity of plant communities. The dominance of this plant can cause the disappearance of other species, especially those that are less competitive. 

Species are important for maintaining ecosystem balance, as different species fulfill various functions, including microclimate regulation [4]. The weed infestation of maize by *V. arvensis* can reduce yields by reducing the light and the nutrient availability required by the crop plant [4,5]. Understanding the ecology and management strategies for *V. arvensis* in maize cultivation is essential for effectively controlling this weed species [6,7]. It is quite common and poses a challenge for farmers in many maize-growing regions. Some biotypes of field pansy can be difficult to control with herbicides without damaging maize plants, which makes it challenging to manage this weed in maize cultivation [8]. The concurrence of field pansy depends on various factors, including temperature, soil moisture, soil type, and competition with other plants [9]. This plant is relatively flexible, but it thrives best in areas that have moderate soil and climatic conditions.

This study’s working hypothesis assumed that the occurrence of *V. arvensis* in sugar maize cultivation depends on agricultural conditions and weather conditions.

The aim of the study was to assess dynamic changes in the status and the degree of *V. arvensis* infestation in sugar maize that was cultivated in the Wielkopolska region in the past three decades.

## 2. Results

The distribution of the percentage mass contribution of field pansy, except when the previous crop was winter wheat, was indicated by short whiskers. The presence of short whiskers indicated that one-fourth of the observations had values of field pansy that were close to the maximum and one-fourth had values that were close to the minimum, while one-half of the observations had a wider range of values of field pansy. When winter wheat was the previous crop, the median was below the upper quartile, indicating a right-skewed distribution. This was further confirmed by the presence of a longer upper whiskers and by outlier observations.

The distribution of the percentage mass contribution of field pansy in the number of weeds after previous crops of winter rapeseed, winter rye, and spring barley was also indicated by short whiskers. However, when the previous crops were winter wheat and winter rye, the median was located below the lower quartile, indicating right-skewedness. On the other hand, if maize was the previous crop, the distribution was left-skewed. Additionally, the upper whiskers were longer after winter wheat. The absence of lower whiskers suggested that observations below the lower quartile did not differ much from the smallest value below the median.

The data presented in Figure 1 show significant variability in the percentage of fresh weight and the number of field pansy in weeds, depending on the previous crops of maize. This variability was likely influenced by numerous favorable or unfavorable factors. Furthermore, Figure 1 demonstrates a lack of conformity to a normal distribution and a lack of homogeneity in the variance of the mass and number of field pansy in the overall weed population after using different previous crops of maize.

### 2.1. The Impact of Meteorological Conditions on Field Pansy Weed Infestation

A correlation analysis was conducted to determine the impact of meteorological conditions on the percentage contribution of field pansy in the total weed weight and the percentage contribution of field pansy in the total weed count (Table 1). A significant negative correlation was found between the percentage contribution of field pansy in the total weed weight and air temperature in June (i.e., lower air temperature in June resulted in an increased percentage contribution of field pansy). Rainfall did not have any impact on the field pansy contribution. For the variable on the field pansy contribution in the total weed count, all correlations were insignificant. Most correlation coefficients were close zero, indicating their negligible (random) influence on the field pansy contribution to the overall weed mass.

Despite the weak individual relationship, an investigation was conducted to determine whether a multiple regression model (with independent variables of rainfall and temperature) could sufficiently explain the variability in the percentage contribution of field pansy in the total weed weight. The dependent variable was transformed using the natural logarithm to achieve a normal distribution. As mentioned in the description of the statistical methods, it was not possible to obtain a normal distribution for the second variable (the field pansy contribution in the total weed count). Additionally, the Spearman rank correlation coefficients were even smaller than those for the field pansy mass (Table 1), which is why a regression analysis was not performed for the variable.

Before proceeding with the analysis, Spearman rank correlation coefficients were calculated between the explanatory variables (Table 2) to exclude highly correlated independent variables. The variables with the strongest correlation with the dependent variable (Table 1) were selected for the model, as well as variables that were strongly correlated with each other (Table 2). Since significant correlations were found between rainfall in May and June, as well temperature in May and June, the variables selected for the model were rainfall in April and May and temperature in April and June (these variables were also the most strongly correlated, although it should be noted that most of these correlations were not significant, and they were correlated with the dependent variable).

The forecasting equation obtained through stepwise multiple regression analysis revealed a significant relationship between the proportion of field pansy in the total weed mass and the temperature in April and June, while it should have had a statistically insignificant relationship with rainfall in May.
$$y=\exp\left(3.3767^{*}+{\;0.1711^{*}t}_{k}-{0.4876^{*}t}_{cz}+0.0066o_{m}\right)$$
where *y* represents the proportion of field pansy in the total weed mass, ${\;t}_{k},\;t_{cz}$ represents the temperature in April and June, respectively, $o_{m}$ represents the rainfall in May (asterisks denote statistical significance at the 0.05 significance level), and the coefficient of determination R^2^ is 0.291, indicating that the regression equation explains only 29% of the total variability in the future y. The residuals in the model follow a normal distribution (W = 0.9832, *p* = 0.2133).

When treating the variable “proportion of field pansy in the total number of field pansy” as a dichotomous variable, obtained through stepwise logistic regression (where none of the variables were removed from the model), the logistic regression model predicting the dependency of the number of field pansy individuals on temperature and rainfall is as follows:(1)$$p=\frac{e^{\beta X}}{1+e^{\beta X}},$$
where:$$\beta X=-3.0084^{*}-0.2206^{*}{\;t}_{k}-0.3076^{*}t_{cz}+0.0128^{*}o_{k}-0.0078^{*}o_{m},$$
p represents the probability of field pansy occurrence, the symbols ${\;t}_{k},t_{cz},o_{m}$ are the same as in the forecasting equation, and $o_{k}$ represents rainfall in April.

### 2.2. The Impact of Categorized Meteorological Conditions on Field Pansy Weed Infestation

Because temperature and rainfall had a slight impact on the variables under investigation, the year was categorized based on humidity and temperature (Table 3). The relationship between the proportion of field pansy mass in the total weed mass and the categorized variables (temperature and rainfall) was analyzed using the non-parametric Kruskall–Wallis test (the assumption was that the variance of homogeneity was not met—in Lewene’s test for rainfall, *p* = 0.0025; for temperature, *p* = 0.0031). The hypothesis of no significant difference was rejected at the 0.05 significance level because the test statistic (*H* = 46.764) significantly exceeded the critical value and the *p* value was *p* = 0.0000. The non-parametric multiple comparisons test identified three overlapping homogenous groups (Table 3), where one group combined dry years (D) with very wet (VW) and extremely wet (EW) years. Dry years (D) significantly differed from average (A) and wet (W) years, as well as from very dry (VD) and extremely dry (ED) years. Similarly, very wet (VW) and extremely wet (EW) years significantly differed from wet (W) and Wet (W) and Average (A) years significantly differed from Wet (W) years.

Next, a similar relationship was examined in terms of categorized temperature. It was found that the proportion of field pansy in the weed mass significantly differed, depending on the prevailing temperature in the given years (*H* = 9.5951 *p* = 0.0083). A further detailed test (Table 4) showed that moderate (M) years significantly differed from cold (C) years.

A similar analysis was conducted for the proportion of field pansy in the total number of weeds. Due to the lack of normal distribution in the analyzed variable (including the transformed variable, as mentioned in the statistics discussion above), a non-parametric Kruskal–Wallis test was performed. The null hypothesis, of no differences between the categories, was rejected for humidity (*H* = 34.594, *p* = 0.000). However, there was no evidence supporting the rejection of the null hypothesis of no difference in the proportion of field pansy in the total number of field pansy during the warm, cold, and moderate years. The non-parametric multiple comparison test for temperature categories indicated the extremely dry (ED) years significantly differed from the dry (D) and very dry (VD) years (Table 5).

A factorial analysis of variance for the proportion of field pansy in the total number of weeds was conducted using the non-parametric Kruskal–Wallis test. The years significantly differed from each other (*H* = 99.908, *p* = 0.000). The multiple-comparison tests (Table 6) revealed that the year 2019 significantly differed from 2005, 2008, 2011, and 2004. Additionally, 2018, 2017, and 2003 significantly differed from 2011 and 2004, while 1995 differed significantly from 2004. The remaining years did not show significant differences. As observed, the homogenous groups encompassed all temperature and humidity categories.

### 2.3. Analysis of the Relationship between the Four Crop Field Pansy Weed Infestations

A non-parametric analysis of variance using the Kruskal–Wallis test was conducted (i.e., a non-parametric test, due to violation of the assumption of homogeneity of the variances *p* = 0.0022) for the dependent variable of the proportion of field pansy in the total weed mass, based on the type of previous crop. Significant differences were found between the levels of the factor (*H* = 14.21; *p* = 0.0143). Multiple-comparison tests revealed that the proportion of field pansy in maize planted after maize significantly differed from the proportion of field pansy in maize planted after winter wheat (Table 7).

A similar analysis was conducted for the participation of field pansy individuals in the over number of weeds. Significant differences were found between the levels of the factor (*H* = 14.71; *p* = 0.0117). However, the non-parametric test of multiple comparisons did not show any differences between the types of previous crops.

### 2.4. Analysis of Weed Infestation Depending on the Previous Crops and Weather Conditions

An analysis of the dependent variable, the participation of field pansy in the total number of weeds, was conducted based on the preceding approach and the meteorological conditions, using the generalized linear model with a logit link function. For this purpose, the dependent variable was presented in dichotomous form, with each weed defined as 1 (field pansy) or 0 (other weed). Significant collinear variables were excluded from the analysis, and a stepwise method was applied. All variables were entered into the model. The conducted analysis showed that the abundance of field pansy individuals was influenced by the temperature, precipitation, and the previous crop (Table 8).

The logistic probability model for the occurrence of field pansy was as follows:(2)$$p\;\left(previous\;crop\right)=\frac{e^{\beta X}}{1+e^{\beta X}}\;,$$
where
$$X=-2.9697^{*}-0.5472m-0.8351^{*}ww+1.6944^{*}wt-0.2440rz\;+0.1361wr-0.1709^{*}t_{k}+0.3250^{*}t_{cz}+0.0156^{*}o_{k}+0.0037^{*}o_{m},\;\text{and}$$
*m* represents maize, *ww* represents winter wheat, *wt* represents winter triticale, *rz* represents winter rape, *wr* represents winter rye, and other symbols mean the same as in Equation (1)

The participation of field pansy in the total number of weeds differed significantly when the previous crop was winter triticale (Table 9). The probability of field pansy occurrence determined from the model was very low (Figure 1 shows that all observations—the proportion of field pansy individuals in the total number of weeds—did not exceed value 0.1). The highest probability of field pansy participation in the total number of weeds was obtained when the previous crop was winter wheat, but the value did not differ significantly from the probability when maize was used as the previous crop. The participation of field pansy in the total number of weeds did not differ significantly when the previous crop was winter rapeseed, spring barley, winter rye, or maize.

## 3. Discussion

This study aimed to understand the impact of air temperature and soil moisture on the infestation of sugar maize by field pansy. The results indicate that the lower air temperature in June had a significant influence in increasing the participation of field pansy in the fresh weight of weeds found in maize. A higher fresh weight of field pansy was observed in years when April had a higher air temperature and when May had an increased rainfall. According to Dobrzański [10], field pansy is a species that germinates at a minimum temperature ranging from 2 °C to 7 °C, with an optimum temperature of 12 °C to 13 °C, and its maximum germination temperature can reach 30 °C to 35 °C. Dobrzański stated that temperature in an important factor, as well as the dynamics of weed population changes. Consequently, the species composition and the degree of weed infestation in the same crop can vary significantly from year to year, primarily due to changes in thermal and moisture conditions during specific growing seasons.

The moisture of the soil had the greatest impact on increasing the number of field pansy among the overall weed population.

The results of our research showed that the probability of a higher mass of field pansy to the overall weed biomass was highest when maize was sown after winter wheat. These findings were consistent with those of previous studies [11], in which field pansy individuals were more frequent when sugar maize was sown after winter wheat. On the other hand, the lowest probability of a higher density of field pansy individuals occurred when maize was sown after winter rye. Fired et al. [12] suggested that the species composition of weed in a particular area was influenced by the type of cultivated plant, the previous crop, the soil type, and pH, as well as by the geographical region and climate. On the other hand, greater species-composition diversity of weeds was observed in plantations where crops were grown in rotations rather than in monocultures [13,14,15]. Those authors also claimed that changes in weed species composition occurred as a result of simplifying the cultivation method of tillage [16]. 

Chovancowa et al. [17] found significant differences in the spectrum of weed species when maize was grown in monoculture, depending on the cultivation method before sowing. The main differences were related to plowing cultivation, compared to conservation tillage methods. Chovancowa et al. found that reducing the depth of tillage led to an increase in the proportion of perennial weed infestation in maize. On the other hand, conservation tillage, especially no tillage, led to an increase in the proportion of persistent species. Torresen et al. [18] believed that performing traditional plowing evenly distributed weed seeds in the cultivated soil layer, while minimum tillage concentrated the seeds in the upper soil layer, from which they could easily and more intensively germinate. Conversely, Vakali et al. [19] argued that simplified cultivation methods could cause changes in the weed-species spectrum in the community. Plowing, for example, displaces weed seeds to greater depths, where their germination is inhibited, thus contributing to a change in the weed-species composition in the community.

## 4. Materials and Methods

This research was conducted during the years 1992 to 2019 in the fields of the Research and Education Center Gorzyń, Złotniki branch (52°29′ N, 16°49′ E), which belongs to the Poznań University of Life Sciences. Sugar maize was sown annually after various other preceding crops—winter triticale, winter wheat, winter rapeseed, spring barley, winter rye, and maize grown for grain harvest. The evaluation of sugar maize weed infestation was carried out in experiments related to chemical weed control in maize, which were established as single-factor randomized block designs in four field replications. The study considered data about weed infestation in control plots where no herbicide treatments were applied. The experimental plots had an area of 11.8 m^2^ and consisted of four rows of maize plants. The row spacing was 70 cm and the plant spacing within the rows was 25 cm, resulting in 24 plants per row. Maize seeds were sown manually with 2 grains per point. At the second or third leaf stage of maize (BBCH 12-13), thinning was performed to leave only one plant per point. The assessment of the condition and degree of weed infestation (i.e., the number and the fresh weight of the weeds) in the control plots was carried out annually in late June and July. The evaluation involved placing frames with dimensions of 0.7 × 0.5 m in randomly selected locations within each plot, identifying and determining the number of all weed species that presented within the designated areas. After removing all weeds from the specified area, they were sorted into individual species, counted, and weighed.

Phytosociological analysis was conducted on fixed research plots, where the condition and degree of weed infestation (i.e., species abundance and fresh weight) of sugar maize were determined [20]. The individual weed species were classified according to the phytosociological system developed by Matuszkiewicz et al. [21] and their participation was determined throughout the study years.

### 4.1. Statistical Analysis

In the first two years of the study, the presence of the field pansy (*Viola arvensis*) was not observed, so those years were excluded from the analysis. The field pansy first appeared in the third year of the study (1994).

A box plot (i.e., a box-and-whisker plot) and Tukey’s test [22] were used to present the measurement results and the empirical distribution on the field pansy contribution in terms of total weight and number of weeds. The box plot showed the upper and lower quartiles (the box), the median (a point within the box), the range of non-outliers, and any outlier observation (located outside the box by more than 1.5 times the interquartile range); additionally, the maximum and minimum observation could be read, which represented either the most extreme outlier or the end of the range of non-outliers.

The assumption that the variable’s distribution is normal is a requirement in many statistical methods. The assumption was tested using the Shapiro–Wilk test. The variable the field pansy contribution in terms of total weight did not meet the assumption (*W* = 0.7170; *p* = 0.000), so it was transformed using the natural logarithm. The hypothesis of normality for the transformed variable was not rejected (*W* = 0.9788 and *p* = 0.0943). Similarly, the variable field pansy contribution, in terms of total number, did not satisfy the assumption of normality (*W =* 0.8548, *p* = 0.000). After applying the logarithmic transformation, the distribution still adhered to the normal distribution (*W* = 0.9365, *p* = 0.0001).

Therefore, a Box-Cox transformation was performed, but it did not yield the expected result. Another presumption required for the analysis of variance is the homogeneity of variances across compared groups. In the case of non-compliance with these assumptions (normality and homogeneity), the analysis of variance using the Kruskal–Wallis method was employed.

To determine the relationship between the field pansy contribution in terms of total weight and number of weeds and weather conditions (precipitation and temperature), Spearman’s rank correlation coefficients were calculated. The stepwise backward multiple regression method was applied to establish a predictive equation for the field pansy contribution in weed weight, based on meteorological conditions. The collinearity of variables was assessed using Spearman’s variable correlation coefficients (however, the stepwise method did not guarantee the exclusion of collinear variables from the model). A similar analysis could not be conducted for the field pansy contribution in terms of the total number of weeds, due to the lack of a normal distribution. However, each weed species was assigned a value of 1 (if it was the field pansy) or 0 (for the other plants). Subsequently, a stepwise backward logistic regression analysis with a logit link function was performed to obtain a probability forecast of field pansy occurrence, based on meteorological conditions. The relationship between the field pansy contribution in terms of weight and number and the categorized meteorological conditions was examined. The non-parametric Kruskal–Wallis test was employed (since the assumption of ANOVA was not met).

To determine if the study years and the preceding crops significantly differed in terms of the field pansy contribution in the total weed weight (transformed using the natural logarithm of the dependent variable), a one way analysis of variance (ANOVA) model was applied separately for each factor. A two-way ANOVA was not used, due to the non-orthogonality of the design. If the null hypothesis of no difference between factor levels was rejected, simultaneous multiple comparisons using Tukey’s method were employed. For the study years, the assumption of homogeneity of variance was checked using Levene’s test (*F* = 1.56 *df* = 25.78; *p* = 0.0713). Since the assumption of homogeneity of variance was not met for the preceding crops (*F* = 4.0276 *df* = 5.98; *p* = 0.0023), the non-parametric Kruskal–Wallis test was used. Similar analyses (i.e., comparison of the study years and the previous crops) were conducted for the field pansy contribution in the total weed count, using the Kruskal–Wallis method (due to non-normality assumptions).

Using the dichotomous nature of the future “presence of the field pansy” in the total number of weeds (1 represented field pansy, 0 represented other species), a model was developed to predict the probability of field pansy occurring, based on the preceding crop and meteorological conditions. Multiple comparisons were conducted, using the Holm correction [23].

Unfortunately, for the variable proportion of the field pansy in the total weed weight it was not possible to establish a prediction equation based on both previous crop and meteorological conditions, because of the assumption of variance homogeneity was not met (there was no appropriate non-parametric test for covariance analysis).

For all analyses, a significance level of 0.05 was adopted. The statistical software package used was Statistica 13.3together with R environment package utilized stats (R Core team) [24,25,26,27].

### 4.2. Weather Conditions

The descriptions of the weather conditions in each year of the research were based on the measurements conducted at the Meteorological Station near an experimental field in Złotniki that belongs to the Agronomy Department at the Poznań University of Life Sciences. The periods of April, May, and June were considered, and the percentage of precipitation compared to the long-term average was used to characterize the period as extremely wet, very wet, wet, average, dry, very dry, or extremely dry, according to the values provided in Table 10 [28].

The meteorological conditions during the growing season in years from 1992 to 2019, both in terms of temperature and humidity, are shown in Table 11. According to the criteria mentioned above, the years 1999, 2009, 2012, and 2013 were classified as extremely humid, while the year 2010 was classified as very humid. The years 1993, 1995, 2014, 2016, and 2017 were categorized as humid. The years 1994, 1997, 1998, 2006, 2007, 2015, and 2019 were classified as average. The years 2001, 2002, 2004, and 2005 were considered dry. The years 1996, 2000, 2003, 2008, and 2011 were categorized as very dry, and the years 1992 and 2018 were classified as extremely dry. Particularly favorable thermal conditions for the development of plants, especially those requiring warmth, were observed in the years 1992, 1993, 1998, 2000, 2002, 2003, 2006, 2007, 2008, 2009, and 2011, which were characterized as warm years. The years 1994, 1995, 1996, 1999, 2001, 2004, 2005, and 2011 were considered moderate in terms of temperature. The year 1997 was classified as a cold year.

### 4.3. Soil Conditions

The Research and Education Center in Złotniki is located on a glacial outwash plain within the Poznań upland region, which is characterized by the granulometric composition of light clays. According to the PTG classification system [SGP 2011], the soil on which the experiments were conducted can be characterized as follows:Order: fluvisolsSuborder: brown soilsGreat group: typical brown soilsSubgroup: loamy sandsFamily: medium loamy sand, lying shallow on light clay

Throughout the years of research, the soil was classified as bonitation class IV and/or IV b, belonging to the good rye complex.

## 5. Conclusions

The lower air temperature in June resulted in a significant increase in the proportion of field pansy in the fresh weed mass, while the warmer weather in April and May favored a higher proportion of field pansy mass. The meteorological condition had a small impact on the proportion of field pansy in the total weed mass. In the regression model with multiple explanatory variables, only slightly less than 30% of the variability in the field pansy’s mass proportion was explained by meteorological conditions, while over 60% of the variability remained random. The projected average probability of field pansy occurrence in the studied experiment was 0.26 and depended on the temperature in April and June, as well as on the precipitation in April and May. The smallest proportion of field pansy fresh weight was expected in wet and cool years, while the largest proportion was expected in dry and moderate years. The proportion of the number of field pansies in the total number of weeds depended solely on humidity—i.e., the smallest proportion occurred in extremely dry and wet years, while the largest proportion occurred in dry and very dry years. The probability of a higher proportion of field pansy fresh weight in the total weed fresh weight was greater when maize was grown after wheat rather than after maize. The probability of a higher number of field pansy individuals in the total number of weeds was highest when maize was grown after winter wheat, but did not differ significantly from the proportion of field pansy number after maize, and the lowest probability of field pansy number occurred in the total number of weeds was observed after winter triticale.

## Figures and Tables

**Figure 1 plants-12-03581-f001:**
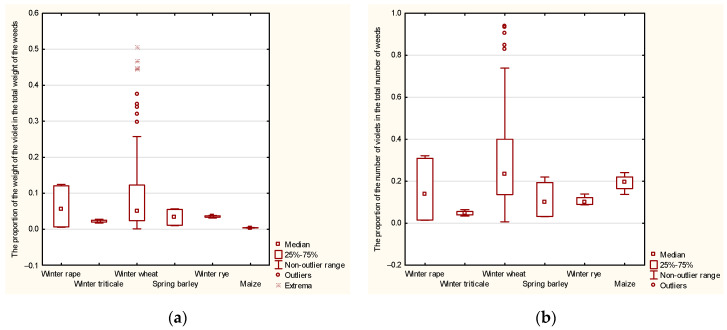
Box plot representing the proportion of field pansy in (**a**) the total mass and (**b**) the number of weeds.

**Table 1 plants-12-03581-t001:** Spearman rank correlations between the percentage contribution of field pansy in the total fresh weed weight and total weed count, rainfall, and temperature during the period from April to June.

Variable	Rainfall				Temperature			
	April	May	June	Sum	April	May	June	Average
Proportion of field pansy in the total weed mass	−0.087	0.154	0.006	0.062	0.108	−0.065	−0.462	−0.203
Proportion of field pansy in the total number of weeds	−0.067	−0.035	−0.053	−0.094	0.289	−0.016	−0.234	−0.007

**Table 2 plants-12-03581-t002:** Spearman rank correlations between meteorological conditions from April to June.

Variable		Rainfall				Temperature			
		April	May	June	Sum	April	May	June	Average
Rainfall	April	1	−0.312	−0.093	0.125	−0.120	0.078	−0.135	−0.166
	May	−0.312	1	0.197	0.676	0.015	−0.459	−0.118	−0.190
	June	−0.093	0.197	1	0.735	−0.215	0.065	−0.395	−0.265
	Sum	0.125	0.676	0.735	1	−0.210	−0.283	−0.36	−0.387
Temperature	April	−0.120	0.015	−0.215	−0.210	1	0.300	0.362	0.758
	May	0.078	−0.459	0.065	−0.283	0.300	1	0.416	0.687
	June	−0.135	−0.118	−0.395	−0.36	0.362	0.416	1	0.769
	Mean	−0.166	−0.190	−0.265	−0.387	0.758	0.687	0.769	1

**Table 3 plants-12-03581-t003:** Homogenous groups indicate that the proportion of field pansy mass in the total weed mass varies significantly, depending of the humidity categories of the years, with the highest proportions observed in the extremely wet and very wet years, intermediate proportions in the average years, and the lowest proportions in the dry, very dry, and extremely dry years.

Humidity Category	D	VW	EW	A	VD	ED	W
Rangs average	81.35	75	64.38	49.57	43.5	27	19.44
Average proportion	0.203	0.100	0.102	0.052	0.048	0.016	0.011
Homogenous group	a *	a	a				
		b	b	b	b	b	
					c	c	c

* The same letters indicate homogenous groups (no statistically significant difference). EW—extremely wet; VW—very wet; W—wet; A—average; D—dry; VD—very dry; ED—extremely dry.

**Table 4 plants-12-03581-t004:** Homogenous groups for the Kruskal–Wallis test on the field pansy mass proportion in the total weed mass, based on the temperature categories of the years.

Temperature Category	M	C	W
Range average	63.73	56.76	44.73
Average	0.129	0.049	0.063
Homogenous group	a *	a	
		b	b

* The same letters indicate homogenous groups (no statistically significant difference). W—warm; M—moderate; C—cold.

**Table 5 plants-12-03581-t005:** Homogenous groups for the Kruskal–Wallis test of the proportions of field pansy in the total number of weed—humidity categories of the years.

Humidity Category	ED	W	A	EW	VW	D	VD
Range average	12.375	23.031	50.018	55.219	67.000	68.650	69.813
Average proportion	0.030	0.073	0.211	0.228	0.257	0.384	0.386
Homogenous group	a *	a	a	a	a		
			b	b	b	b	b

* The same letters indicate homogenous groups (no statistically significant difference). EW—extremely wet; VW—very wet; W—wet; A—average; D—dry; VD—very dry; ED—extremely dry.

**Table 6 plants-12-03581-t006:** Multiple simultaneous comparisons using Tukey’s test. Homogenous group of years based on the dependent variable of the proportion field pansy in the total number of weeds.

Year	Range Average	Average	Humidity Categories	Temperature Categories	Homogenous Group			
2019	3.50	0.115	A	W	a *			
2016	5.50	0.125	W	W	a			
2018	12.38	0.174	ED	W	a	b		
2017	15.00	0.183	W	M	a	b		
2003	17.75	0.200	VD	W	a	b		
1995	20.88	0.218	W	M	a	b	c	
1999	27.50	0.296	EW	M	a	b	c	d
2013	30.88	0.324	EW	W	a	b	c	d
2001	35.25	0.368	D	M	a	b	c	d
2006	41.63	0.398	A	W	a	b	c	d
1997	45.75	0.431	A	C	a	b	c	d
2007	50.00	0.440	A	W	a	b	c	d
1998	50.75	0.445	A	W	a	b	c	d
2014	50.75	0.437	W	W	a	b	c	d
1996	54.25	0.454	VD	M	a	b	c	d
2002	62.00	0.487	DS	W	a	b	c	d
2010	67.00	0.506	VW	M	a	b	c	d
2000	70.50	0.520	VD	W	a	b	c	d
1994	74.38	0.547	W	M	a	b	c	d
2012	79.75	0.588	EW	W	a	b	c	d
2009	82.75	0.608	EW	W	a	b	c	d
2015	84.13	0.644	A	M	a	b	c	d
2005	89.50	0.661	D	M		b	c	d
2008	92.25	0.699	VD	W		b	c	d
2011	98.75	0.861	VD	W			c	d
2004	102.25	0.949	D	M				d

* The same letters indicate homogenous groups (no statistically significant difference). EW—extremely wet; VW—very wet; W—wet; A—average; D—dry; VD—very dry; ED—extremely dry; W—warm; M—moderate; C—cold.

**Table 7 plants-12-03581-t007:** Homogenous group for the participation of the field pansy in the overall fresh weight of weeds, depending on the previous crop from maize, based on the conducted Kruskal–Wallis analysis of variance.

Previous Crop	Maize	Winter Triticale	Spring Barley	Winter Rye	Winter Rapeseed	Winter Wheat
Range average	7.75	32.75	42.25	44.25	48.75	57.80
Average	0.004	0.023	0.033	0.035	0.062	0.107
Homogenous group		a *	a	a	a	a
	b	b	b	b	b	

* The same letters indicate homogenous groups (no statistically significant difference).

**Table 8 plants-12-03581-t008:** The logistic model of the participation of field pansy individuals in the total number of weeds.

	Source of Variation	LR Chi sq.	Df	Pr (>Chi sq.)
	Previous Crop	228.86	5	0.0000
Temperature	April	74.54	1	0.0000
	June	334.44	1	0.0000
Rainfall	April	160.37	1	0.0000
	May	9.97	1	0.0016

**Table 9 plants-12-03581-t009:** Multiple comparisons of preceding crops with Holm’s corrections.

Previous Crop	Winter Triticale	Winter Wheat	Winter Rapeseed	Spring Barley	Winter Rye	Maize
The probability of the presence of field pansy in the total number of weeds determined from model 2	0.05	0.23	0.16	0.12	0.11	0.19
Homogeneous groups		a *				a
			b	b	b	b

* The same letters indicate homogenous groups (no statistically significant difference).

**Table 10 plants-12-03581-t010:** Percentage of precipitation compared to the long-term average.

Type	% Participation
Extremely wet	>200
Very wet	151–200
Wet	126–150
Average	75–125
Dry	50–74
Very dry	25–49
Extremely dry	<25

**Table 11 plants-12-03581-t011:** Temperature and precipitation during the years of research at Research and Education Center Złotniki.

Precipitation [mm]						Air Temperature [°C]				
Year	April	May	June	Sum	Rating *	April	May	June	Average	Rating **
1992	20.1	37.1	3.0	60.2	SS	7.6	13.9	19.3	13.6	C
1993	8.6	86.6	80.4	175.6	W	9.6	16.4	15.0	13.6	C
1994	47.5	66.4	34.3	148.2	P	9.1	12.0	15.9	12.3	U
1995	12.0	77.6	89.1	178.7	W	8.2	12.7	16.0	12.3	U
1996	13.8	74.2	33.8	121.8	BS	8.3	12.7	16.4	12.5	U
1997	39.9	67.6	47.4	154.9	P	5.5	13.0	17.0	11.8	Z
1998	30.4	30.0	80.3	140.7	P	10.4	14.9	17.4	14.2	C
1999	73.6	55.6	88.3	217.5	SW	9.6	13.5	16.5	13.2	U
2000	15.7	47.4	29.9	93.0	BS	12.1	15.7	17.5	15.1	C
2001	33.1	10.4	67.8	111.3	S	8.3	15.2	15.3	12.9	U
2002	34.2	45.7	38.1	118.0	S	8.9	16.8	18.1	14.6	C
2003	16.2	24.0	40.4	80.6	BS	8.6	15.7	19.2	14.5	C
2004	19.4	49.8	51.3	120.5	S	9.7	12.9	16.1	12.9	U
2005	14.5	74.3	19.1	107.9	S	9.4	13.3	16.5	13.1	U
2006	43.6	57.4	26.9	127.9	P	8.8	13.8	18.7	13.8	C
2007	9.3	77.0	59.6	145.9	P	10.8	15.2	19.3	15.1	C
2008	77.5	9.5	8.4	95.4	BS	10.0	16.2	20.6	15.6	C
2009	19.2	109.9	113.8	242.9	SW	12.9	14.0	16.0	14.3	C
2010	26.8	110.5	43.4	180.7	BW	9.3	12.2	18.4	13.3	U
2011	9.8	22.5	66.5	98.8	BS	12.4	15.5	19.9	15.9	C
2012	17.4	84.4	118.1	219.9	SW	9.3	16.3	17.0	14.2	C
2013	10.5	95.5	114.9	220.9	SW	8.9	15.6	18.4	14.3	C
2014	50.3	80.7	44.6	175.6	W	11.4	14.6	17.9	14.6	C
2015	18.4	36.2	82.0	136.6	P	8.3	12.9	16.2	12.5	U
2016	37.4	43.0	83.6	164.0	W	8.6	15.4	18.3	14.1	C
2017	40.6	56.8	68.2	165.6	W	7.6	13.7	17.4	12.9	U
2018	36.2	17.4	25.6	79.2	SS	13.3	16.9	19.1	16.4	C
2019	8.6	94.4	7.2	110.2	P	10.5	11.9	22.0	14.8	C

* EW—extremely wet; VW—very wet; W—wet; A—average; D—dry; VD—very dry; ED—extremely dry. ** W—warm; M—moderate; C—cold.

## Data Availability

Available upon reasonable request.

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
