# Peer review of "The Dynamics of Sugar Maize (Zea mays saccharata Sturt.) Infestation of Field Pansy (Viola arvensis)"

_plants, 2023, doi:10.3390/plants12203581_

Round 1
Reviewer 1 Report
Dear authors,
Your article is interesting. Some remarks in file.
Best regards

Author Response
Reviewer 1
In the introduction you indicate that the purpose of the study is to evaluate the effects of previous crops on the suppression of maize growth by field pansy. In the Materials and methods, you do not mention the use of the previous crops. How did you verify the previous crops before proceeding with the experiments? Without that information, the information has no merit. The use of marginal statistics to find some level of significance is not good science. Most of the results show no significance.
Comments on the Quality of English Language
Minor English editing needed. The larger problem is use of unknown statistical methods. Most readers will have no idea what the results indicate.
Answers:
- Sweet maize over course of 28 years of research (1992-2019) was cultivated after various preceding crops: corn sown for grain harvest, winter wheat, spring barley, winter rye, winter rapeseed and winter wheat (table 9). Sweet maize after each of these preceding crops was grown according to established agricultural practices for these preceding crops was grown according to established agricultural practices for this plant. Indeed, this was not explained in detail in the research methodology. This has now been supplemented. We believe that the lack of significant differences between the analyzed research result is also valuable information for both the reader and agricultural practice. (Thank you for these valuable comments).
- In the research, sweet maize was randomly cultivated after various preceding crops. In addition to assessing the impact of environmental conditions, one of the objectives was determine if the preceding crop had any influence on the condition and level of weed infestation in sweet maize.
The statistical methods employed were chosen based on the assumptions that the analyzed data met. The only assumption of the non-parametric Kruskal-Wallis analysis of variance is that the analyzed data must be measurable, and this assumption was met by the variable under consideration. Therefore, this statistical method was applied. This test is used when the assumptions of parametric tests such as ANOVA are not met, and it examines the effect of one independent variable on a dependent variable across three or more groups. It is often used in such studies. Below, we provide several publications in which similar statistical tests were also applied:
Jaipolsaen, N., Sangsritavong, S., Uengwetwanit, T., Angthong, P., Plengvidhya, V., Rungrassamee, W., & Yammuenart, S. (2022). Comparison of the Effects of Microbial Inoculants on Fermentation Quality and Microbiota in Napier Grass (Pennisetum purpureum) and Corn (Zea mays L.) Silage. Frontiers in Microbiology, 12(2021), https://doi.org/10.3389/fmicb.2021.784535
Valujeva K., Pilecka-Ulcugaceva J., Skiste O., Liepa S., Lagzdins A., Grinfelde I. Soil tillage and agricultural crops affect greenhouse gas emissions from Cambic Calcisol in a temperate climate. Pages 835-846. Received 03 Apr 2022, Accepted 29 Jun 2022, Published online: 11 Jul 2022.
- H. Bloom, D. M. Bauer, A. Kaminski, I. Kaplan, Z. Szendrei, "Socioecological Factors and Farmer Perceptions Impacting Pesticide Use and Pollinator Conservation on Cucurbit Farms," Front. Sustain. Food Syst., vol. 5, p. 672981, Aug. 2021.
Xu Z., Zhang T., Wang S., Wang Z. Soil pH and C/N ratio determines spatial variations in soil microbial communities and enzymatic activities of the agricultural ecosystems in Northeast China: Jilin Province case. Appl. Soil Ecol., 2020, 155, 103629.

Reviewer 2 Report
In the introduction you indicate that the purpose of the study is to evaluate the effects of previous crops on the suppression of maize growth by field pansy. In the Materials and methods, you do not mention the use of the previous crops. How did you verify the previous crops before proceeding with the experiments? Without that information, the information has no merit. The use of marginal statistics to find some level of significance is not good science. Most of the results show no significance.
Minor English editing needed. The larger problem is use of unknown statistical methods. Most readers will have no idea what the results indicate.
Author Response
Reviewer 2
1.Why field pansy? I think in Introduction section it could be said a few words more clear about problem.
- It is not clear how this trial was made. If it was in the field - what about other weeds? Or dominating was field pansy? If yes, other question: why?
3.In the summary is presented that trial was made 1992 - 2002, but later we can found information from 2003 – 2020 as well. What about historical data? Maybe it would be interesting.
- In results section comments are quite strange. I think it should be more talking about results, not how looks figures
Interesting job was done with different statistical methods.
Answers
- The field pansy belong to the group of weeds that have become particularly noticeable in sugar maize cultivation in recent years. Apart from your other weed species, the field pansy has been found to be a prevalent species in sugar maize for most of the years, with a particularly high intensity. Due to the significant presence of this species in the weed population the authors of the publication wanted to illustrate how its contribution evolved over a longer period of time (28 years of research), regardless of environmental conditions. In the introduction, it was also mentioned that effective control of resistant biotypes of the field pansy with commonly used herbicides in sugar maize can sometimes lead to damage in certain varieties of sugar maize plants.
- In the research, the results regarding the presence of the field pansy were based on data collected independently of the preceding crops (several preceding crops). However, the study aimed to determine the occurrence of this weed species based on the weed seed bank accumulated in the soil over an extended period. The field pansy belonged to the group of species with significant seed bank presence, which is way it became the subject of the current publication. The presentation of the contribution of other weed species to the weed infestation in sugar maize, which was also analyzed in the study, will be the subject of subsequent publications.
- In the summary, it was mistakenly stated that the research pertained to the years 1992-2002, whereas it was conducted from 1992 to 2019. This was an editorial error, Therefore, the research results indeed cover a period of 28 years of field experimentation (thank you for bringing this to our attention).
- The description of the obtained results mostly corresponds to the data presented in the charts and tables. We aimed to explain step by step the statistical analysis conducted, which encompassed both the weather conditions during individual years of the research (hence they were categorized into years as EW – extremely wed, W – wet, A – average, Vd – very dry, ED – extremely dry), as well as the specific preceding crop on which sugar maize was cultivated.

Reviewer 3 Report
Comments to the Authors
I have reviewed with interest your manuscript entitled „The dynamics of sugar maize (Zea mays saccharata Sturt.) infestation of field pansy (Viola arvensis)” submitted to the future number of Plants.
The results showed changes in the status and degree of field pansy infestation in sweet corn cultivated after different preceding crops in the last three decades in the Wielkopolska region. The statistical analyses have shown that the probability of a higher number of common violet individuals occurring among the total number of weeds is the highest when maize is cultivated after wheat but does not significantly differ from the proportion of field pansy presence in the total number of weeds occurred when maize was sown in a crop rotation after winter triticale.
In my opinion the current version of your manuscript is suitable for publication in Plants, but after small revisions. The quality of the presentation should be improved. In general, manuscript is well written.
There are same grammar mistakes and awkward sentences, that have to be improved.
Some adjustments are suggested to qualify the paper:
Issues include:
The Abstract is written in proper style, but I suggest to add clear aim of the study after few sentences of introduction of Abstract. Moreover the Abstract should not exceed 200 words. Now there is 223 words. In my opinion it can be stay in this version or Authors may a little shorten it.
General comment to the Introduction section: The content of the literature review chapter is related to the research topic. Up-to-date literature references are presented in the manuscript by the author(s).
In the chapter "Materials and Methods", the methodology is adequate.
In the chapter "Results", the results are displayed correctly.
The “Discussion” is informative. Moreover, the Authors attempt to discuss their important results and the rest is a quotation of literature.
The Conclusions are correctly.
I hope that these comments help you to make an improved version of the manuscript.

Author Response
Reviewer 3
- The abstract is written in proper style, but I suggest to add aim of the study after few sentences of introduction of Abstract. Moreover the abstract should not exceed 200 words. Now there is 223 words. In my opinion it can be stay in this version or Authors may a little shorten it.
- General comment to the introduction section. The content of the literature review chapter is related to the research topic. Up-to date literature references are presented in the manuscript by the authors.
Answers
- We have condensed the abstract to 200 words and explained the purpose of the conducted research. We hope that this shortened version of the abstract better conveys the purpose of our research.
- In accordance in the reviewer’s suggestion we have included citations from several references in the introduction of the manuscript.
We appreciate the constructive comments which we hope an improved version of the manuscript.

Round 2
Reviewer 2 Report
The following are my assessments of paper 2540182: The dynamics of sugar maize infestation of field pansy (Viola arvensis). The researchers collected data for over 20 years in fields that were rotated to sweet maize during some years. They were looking for effects of follow-crops on pansy populations. The results indicate that the primary cause of different pansy populations was weather. Their conclusion states: "The smallest proportion of field pansy fresh weight can be expected in a wet and cool year, while the largest proportion can be expected in a dry and moderate year." The rotational crops appeared to have no effect on pansy populations. Most data collected was non-significant. Therefore, it appears to me that no new information is presented in this paper. In my opinion, a paper needs to present new information, or at least repeat previous research that was significant, to be worthy of publication. This paper does not meet that level. There is nothing the authors can do to repeat the experiments or change the results.The following are my assessments of paper 2540182: The dynamics of sugar maize infestation of field pansy (Viola arvensis). The researchers collected data for over 20 years in fields that were rotated to sweet maize during some years. They were looking for effects of follow-crops on pansy populations. The results indicate that the primary cause of different pansy populations was weather. Their conclusion states: "The smallest proportion of field pansy fresh weight can be expected in a wet and cool year, while the largest proportion can be expected in a dry and moderate year." The rotational crops appeared to have no effect on pansy populations. Most data collected was non-significant. Therefore, it appears to me that no new information is presented in this paper. In my opinion, a paper needs to present new information, or at least repeat previous research that was significant, to be worthy of publication. This paper does not meet that level. There is nothing the authors can do to repeat the experiments or change the results.
Author Response
Reviewer 2
The following are my assessments of paper 2540182: The dynamics of sugar maize infestation of field pansy (Viola arvensis). The researchers collected data for over 20 years in fields that were rotated to sweet maize during some years. They were looking for effects of follow-crops on pansy populations. The results indicate that the primary cause of different pansy populations was weather. Their conclusion states: "The smallest proportion of field pansy fresh weight can be expected in a wet and cool year, while the largest proportion can be expected in a dry and moderate year." The rotational crops appeared to have no effect on pansy populations. Most data collected was non-significant. Therefore, it appears to me that no new information is presented in this paper. In my opinion, a paper needs to present new information, or at least repeat previous research that was significant, to be worthy of publication. This paper does not meet that level. There is nothing the authors can do to repeat the experiments or change the results.
Answers
In our opinion, the Reviewer does not fully understand the specifics of long-term research. We disagree with their assertion that if there is no statistical significance in the results of the conducted research, the manuscript does not contribute anything new and should not be published. In our view, the lack of significant differences confirms the absence of an impact of the analyzed factor, which is also a valuable outcome. The manuscript's significant value lies in the fact that the research has been conducted for over 20 years, which is currently a rarity in publications.

Round 3
Reviewer 2 Report
Not much has changed.